# Semantic Distributed Data for Vehicular Networks Using the Inter-Planetary File System

**DOI:** 10.3390/s20226404

**Published:** 2020-11-10

**Authors:** Victor Ortega, Jose F. Monserrat

**Affiliations:** 1Casa Systems, 46014 Valencia, Spain; 2iTEAM Research Institute, Universitat Politècnica de València, 46022 Valencia, Spain

**Keywords:** semantic data, V2X communications, cooperative perception, P2P, ICN, IPFS, QUIC

## Abstract

Vehicular networks provide means to distribute data among intelligent vehicles, increasing their efficiency and the safety of their occupants. While connected to these networks, vehicles have access to various kinds of information shared by other vehicles and road-side units (RSUs). This information includes helpful resources, such as traffic state or remote sensors. An efficient and fast system to get access to this information is important but unproductive if the data are not appropriately structured, accessible, and easy to process. This paper proposes the creation of a semantic distributed network using content-addressed networking and peer-to-peer (P2P) connections. In this open and collaborative network, RSUs and vehicles use ontologies to semantically represent information and facilitate the development of intelligent autonomous agents capable of navigating and processing the shared data. In order to create this P2P network, this paper makes use of the Inter-Planetary File System (IPFS), an open source solution that provides secure, reliable, and efficient content-addressed distributed storage over standard IP networks using the new QUIC protocol. This paper highlights the feasibility of this proposal and compares it with the state-of-the-art. Results show that IPFS is a promising technology that offers a great balance between functionality, performance, and security.

## 1. Introduction

In 2001, Sir Timothy John Berners-Lee, founder of the World Wide Web Consortium (W3C), published his famous article introducing the Semantic Web [1] and his depiction of the future of Internet. He envisioned a world where Web servers publish the data in a format that is comprehensible not only to people but also to machines. Almost two decades later, the Web has diverged from his vision, but the original concepts and the related research are now more relevant than ever in other communication fields. Internet of Things (IoT) devices are said to be one of the main beneficiaries of the application of semantics, as depicted by A. Jara et al. in [2], but the generally limited capabilities and scope of IoT devices do not provide the most favorable playground for developing an autonomous semantic network, instead requiring the help of external processing. However, there is a particular class of autonomous intelligent and connected devices that offers substantial processing capacity and strong connectivity options, and can afford the extra layer of complexity required to work with semantic data. These are the vehicles and road-side units (RSUs) that form parts of vehicular networks. This new field of application opens up multiple options to vehicular networks and brings a new dimension to Berners-Lee’s vision.

Vehicles and RSUs benefit from a wider variety of sensors as compared with a typical IoT device. Parts of these readings are used as direct inputs to take immediate decisions, such as starting emergency braking or turning on the windshield wipers. Sharing raw sensor data, such as vehicle speed or current position, with the rest of nodes of the network is viable but inefficient and unproductive. However, layered over these basic measurements, intelligent vehicles have the capacity to generate derived complex data obtained after processing and combining the basic data from their sensors. For example, computerized vehicles have the ability to detect other vehicles and estimate their trajectories. The processing and optimization of sharing digested data becomes a powerful tool that can be used for multiple functions without requiring immense quantities of real-time data. As an exemplary use case, traffic congestion could be reduced by using route estimations, detection of road hazards, the adjustment of traffic lights, or the exchange of trajectories in cross junctions. Sharing all this complex sensor data without any clear organization would force vehicles to implement overly complicated processing algorithms. The application of the existing semantic data research and W3C recommendations to the vehicular networks has the potential for reducing the computational burden of data processing, and it helps in painting an accurate real-world representation of the status of the vehicles and the roads, simplifying the developing of advanced autonomous systems capable of interacting with people in a natural way.

When the Semantic Web was introduced by the W3C, their main tasks were to create open data standards for Web contents and design intelligent agents. Semantic agents are autonomous software entities that can analyze and perform tasks using the semantic data [3]. A semantic agent can perform a simple search task, such as “Where is the closest gas station?” and more complex tasks, such as “Tell my wife when I will arrive home [based on the current location and traffic state]”. Agents perform these tasks by collecting data from multiple sources. In this paper, we propose a system that enables the usage of intelligent agents in vehicles and RSUs. With this new solution, agents would have access to all the semantic data published by vehicles and RSUs in an efficient and transparent way. The system is based on the Inter-Planetary File System (IPFS) [4], an open source peer-to-peer (P2P) hypermedia solution that includes native support for the storage and distribution of semantic data. Figure 1 depicts a vehicle connecting to the IPFS network and instantaneously having access to all the semantic information provided by RSUs, other vehicles, and any other remote IoT peer connected to the network. Based solely on IPFS, the proposal of this paper is a completely public and distributed network, unmanaged but secure and trustworthy thanks to the redundancy provided by the existence of multiple peers and the innovative methods used to store and share the information. The resultant P2P network brings the potential and advantages of information-centric networking (ICN) [5] to vehicular networks without requiring a network architecture overhaul.

IPFS is multi-platform software that works over standard IP networks. Therefore, it allows RSUs and other IoT devices to interact independently of their connection to the network. Vehicles can use specific wireless systems, such as dedicated short-range communications (DSRC), the European Telecommunications Standards Institute’s (ETSI) intelligent transport systems (ITS) G5 Standard, or the 3rd Generation Partnership Project (3GPP) Cellular Vehicle-To-Everything Standard; and RSUs can make use of multiple interfaces, wireless and wired, to connect simultaneously to vehicles, other RSUs, and IoT devices such as traffic cameras.

### 1.1. Related Work

The term Semantic Web of Things that combines the Semantic Web and IoT devices was coined in 2009 by F. Scioscia and M. Ruta at [6]. This work has been extended by multiple authors since then. For instance, the Semantic Smart Gateway Framework by K. Kotis and A. Katanasovis is remarkable [7], as is the interoperability analysis by A. Gyrard et al. [8]. The recent article by D. Androcec et al. provides an ample review of this area of research [9]. The work with semantic distributed sensors by H. Choi and W. Rhee [10] is especially interesting—research that extends the work by A. Sheth et al. [11]. Besides semantic data, the natural human–vehicle interaction requires vehicles’ proficiency of spatial language [12]. Moreover, the work by M. Cristani and N. Gabrielli reflects the complexity of spatial and temporal reasoning systems [13].

Prior to any mention of the Semantic Web, K. Sohrabi et al. proposed the usage of P2P networks for distributing sensors in [14], and N. Patwari et al. investigated how utilize P2P sensor networks for cooperative localization in [15]. Due to the convenience of cloud computing, the fog computing [16] paradigm emerged as an alternative that provides some of the benefits of P2P networks without disregarding the usage of central servers. The recent research by D. Tracey and C. Sreenan [17] delves into the advantages of using a pure P2P approach for fog computing. One of the major impediments to a more widespread use of P2P networks has been the lack of standard protocols and applications. However, in recent years, there has been growing research interest in blockchain technology, a highly popular and promising P2P solution. Blockchains have been studied and applied to IoT and vehicular networks by multiple authors [18,19,20]. Although related to blockchain, IPFS is a distinctive P2P protocol. In fact, IPFS and blockchain are considered complementary technologies [21]. S. Muralidharan and H. Ko researched the feasibility of using IPFS in IoT devices in [22].

This paper offers experimental results of a real implementation of a semantic network created with IPFS and compares them with the simulation results published by X. Wang and Y. Li [23]. These simulation results were obtained using an optimized ICN protocol that provides content-addressed data delivery. The usage of ICN has been considered by several authors as the optimal solution for reducing the usage and latency of IoT [24] and vehicular networks [25]. The aforementioned experiment used IPFS over QUIC, a new Internet protocol whose performance has been already analyzed by P. Megyesi et al. [26]. There are no prior studies, to the best of the authors’ knowledge, evaluating the performance of IPFS or QUIC as content-delivery protocols for vehicular networks.

### 1.2. Outline

This rest of the paper is organized as follows: Section 2 analyzes the usage of semantics and intelligent agents within the context of vehicular networks. Section 3 presents the concept of IPFS while discussing its convenience for sharing semantic data among vehicles and RSUs. Section 4 describes the differences between a traditional P2P network and the proposed P2P network. Section 5 details several usage scenarios and their expected behavior. A brief Section 6 discusses security, integrity, and privacy concerns of the resultant network. Then, Section 7 describes and discusses the experiment results and includes a comparison with state-of-the-art simulations from [23]. Finally, Section 8 draws the main conclusions of this paper while introducing some future lines of research.

## 2. Semantic Data and Ontologies

The semantic representation of the data published by vehicles and RSUs makes possible these agent-based machine-to-machine interactions. For instance, a vehicle circulating in Whitechapel could easily ask the network if the Tower Bridge is now open for traffic. There are endless ways to portray transportation data using semantics, and for this reason, agents and data producers need ontologies to provide coherence and unambiguity to the data. Quoting J. Hendler, one of the originators of the Semantic Web, an ontology is defined as a “set of knowledge terms, including the vocabulary, the semantic interconnections, and some simple rules of inference and logic for some particular topic” [3]. This abstract concept is easier to understand with an example. Figure 2 shows a basic usage of semantic data describing a person using the Friend of a Friend (FOAF) ontology. Semantic data are represented by triples (subject–predicate–object). In this example, the subject is the depicted person, the predicate is indicated with an arrow, and the object is the pointed value. For example, the person’s name is Thomas Alva Edison and this person is the friend of Henry Ford. This generic structure can be used to describe highly complex concepts and their relationships.

In regard to vehicular networks, the ontology or ontologies needed to represent and distribute the data should include numerous details about the different states of the vehicles, RSUs, and their surroundings. For instance, basic details could be the description of how to share the information captured by a traffic camera, or the possible readings of weather stations. Additionally, they should also include semantic definitions about static elements of the vehicular networks, such as city buildings, roads, and traffic rules. These definitions, although not directly used while sharing data, would be needed to establish a common communication framework. A comprehensive ontology would also include specific details, such as the difference between a rotary and a roundabout, how many lanes have the different kinds of roads, and the traffic rule to apply when two vehicles arrive at the same time to an uncontrolled intersection.

The task of defining a single large and consistent ontology for every possible concept and situation in every different city and every particular vehicular network would be a tremendous and nearly impossible task. It is more realistic for researchers and developers to give support to multiple interlinked ontologies, promoting a system with the ability to adapt to current and future changes in ontology definitions. The first published ontology for transportation networks was published by B. Lorenz et al. in 2005 [27], but numerous alternative ontologies have been proposed since then. The recent survey by M. Katsumi and M. Fox in [28] summarized and analyzed 11 different transportation ontologies. The most prevalent and complete ontology is the Km4City ontology [29]. It provides classes for streets, points of interest (POIs), public transport, and more. Besides, vehicular networks do not need only transportation ontologies. Additional ontologies, such as the Car Accident Ontology for VANETs (CAOVA) [30], can be used to describe car accidents. Others, such as the Semantic Sensor Network (SSN) W3C recommendation, can be used to create detailed reports of the available sensors. Figure 3 provides an example of the CAOVA ontology.

It is important to note that the task of adding support for existing or future ontologies should not require changes in the network protocols or infrastructure. Ideally, a system change should only comprise a small software update thanks to its compliance with the Resource Description Framework (RDF) and the Web Ontology Language (OWL) W3C recommendations. RDF specifications are a series of guidelines to be used to represent and work with semantic data, and the W3C does not force a specific syntax or programming language. For this task, W3C offers additional specifications, such as the initial RDF/XML syntax and the recent and more user-friendly JavaScript Object Notation (JSON) syntax. Semantic data generators have to use RDF according to the rules specified by the chosen ontologies. W3C ontologies are described using OWL, a complete and powerful metalanguage that, remarkably, uses RDF to define ontologies. Using an analogy, an ontology written with OWL is like a book about English grammar that has been written using the same English grammar rules that are described in the book.

## 3. IPFS (Inter-Planetary File System)

IPFS is an open-source peer-to-peer (P2P) solution for storing and distributing files, websites, or any other kind of data (e.g., maps) in public or private networks. Originally created by J. Benet [4], IPFS is based on the classic BitTorrent file-sharing protocol, and Git, the industry standard version control system. Despite some similarities, IPFS is not another blockchain network. Like Bitcoin or Ethereum, IPFS provides data immutability: once a node stores new content and shares it with other nodes, it is impossible to delete or modify it. However, IPFS nodes do not use any consensus protocol to add data, and each node only has knowledge about a small fraction of the content stored in the network. For these reasons, IPFS is faster than blockchain and highly scalable. This section analyzes how this technology, designed for breaking the dependence of the Web with centralized servers, can be used to provide edge intelligence to vehicular networks.

IoT and vehicular networks entail a huge challenge in terms of network computing. The premise for developing more services based on edge computing and less based on cloud services is clear: the majority of the content is produced by nodes at the edge of the network, and in terms of network usage and latency, it is more efficient to process it in that same edge of the network [31]. Regarding the vehicular network context, vehicles are end-users making use of edge services running on nearby RSUs. These connections are more direct and efficient than connections to remote cloud servers. However, edge computing solutions are traditionally envisioned as hierarchical: the vehicles connect to the edge services provided by the RSUs, but the RSUs still need to connect to remote servers because most of the data can only be retrieved from them. The usage of a P2P systems alleviates this dependence.

The inconveniences of hierarchical solutions, and how P2P communications present an improvement, can be illustrated with a use case: the interested vehicle sends an information request to a nearby RSU to retrieve data. The expectation of the vehicle is to get all the data in a single, fast, and locally confined transaction. It is likely that the information requested by the vehicle is currently unavailable in the nearby RSU, but available in other RSUs. There are multiple alternative approaches on how to solve this issue and allow the vehicle to retrieve the desired information:
The RSU replies to the vehicle that it is unable to return the requested information. The vehicle, on its own, has to send the same request to a different RSU, with a considerable cost in complexity, time, and bandwidth. This option does not comply with the vehicle expectations.Before replying to the vehicle, the RSU connects to a central server to fetch (search and download) the requested information. This option saves time and bandwidth to the vehicle, but requires a server with access to all the data generated by the RSUs.A faster variant of the previous option: The RSU asks the central server not the information but the network address of the RSU that has the information requested by the interested vehicle. Next, the RSU connects to the other RSU to download the information. This approach reduces the number of messages between the RSU and the server. Most of the messages are exchanged between RSUs that are geographically close to each other.Lastly, the P2P variant of the previous option: There is no central server, the RSU is part of a P2P network, and thanks to a distributed search algorithm, the vehicle can find the location of the information and download it without needing to use any hierarchical infrastructure.


P2P solutions are potentially faster and more efficient than any other option, although they are technically more complex in terms of software. This is the reason why this paper considers the usage of IPFS—mature and fully functional P2P software that could greatly facilitate the creation of a P2P network of vehicles, RSUs, and other IoT devices. IPFS software has been designed with large scalability and fast speed as main characteristics. The distribution and search of content in the P2P network is done using a DHT (distributed hash table) based on the Kademlia algorithm [32]. The DHT allows peers to find content stored in any other peer of the network using only the cryptographic hash of the content. For this task, IPFS supports multiple hash algorithms, although by default it uses 32 bytes SHA-256 hashes. A previous study by J. Falkner et al. demonstrated that the DHT algorithm can be used to find content in a P2P network with one million peers [33]. The DHT search uses a hash as input and the result is a list with the peers that currently hold content matching that hash. The interested peer can then select any peer from the list and establish a direct connection to download the content. As the content search is done by hash, IPFS guarantees that the original content has not been altered. If the content is modified, the hash is effectively changed, and the content is rejected. This is exactly the reason that IPFS does not support searching by content name or description, as it would allow peers to tamper with the content before sending it to others. However, this safety measure becomes an inconvenience for vehicle networks, given that the interested node would need to know the hash of the requested information prior to fetching it from the network. Figure 4 shows the process of sharing a file with other peers. Note from the figure that IPFS 32 bytes SHA-256 hashes are encoded as strings using Base58 and adding the prefix Qm.

Although the example shows how to share a file, IPFS is not file-oriented but block-oriented. Each shared block has a maximum size of 256 kB and can be individually addressed using its hash. Content hashes are known as CIDs (content identifiers). When a large file is shared, IPFS splits it into multiple interlinked blocks. IPFS can also share complex file structures (such as Web pages). The interlinked blocks are stored using a Merkle tree [34], a structure that guarantees tamper proof storage and avoids data duplication. This technical solution is also used by Git version control software. Thanks to Merkle trees, nodes only need to know the CID of the root block of the tree to be able to download the whole structure with complete guarantee of integrity of all the downloaded blocks.

### 3.1. Advantages of Content-Addressing

Although IPFS works over standard IPv4 or IPv6 networks, the resultant P2P network shares many similarities with ICN. The usage of the ICN paradigm has been thoroughly explored by several researchers [25,35]. The key advantages shared by both IPFS and ICN are:
Content-addressing. Applications built over IPFS do content requests by CID and do not need to know the location of the content or its uniform resource identifier (URI).Decentralization. As the content is not tied to a specific host, there is no need for permanent servers and the same piece of content can be retrieved from multiple nodes. When one node is disconnected from the network, the network topology changes automatically, and the content published by the detached node is still available from other nodes.Scalability. Network load is naturally balanced avoiding bottlenecks. If the content published by a node is highly demanded, multiple nodes will have a copy and can provide it to others, effectively distributing the network usage.End-to-end data integrity. IPFS hash-addressing method guarantees that the data are never tampered with.Reduced latency and fault tolerance. As every piece of content is transparently retrieved from the best available location, latency is improved, and if the source unexpectedly becomes unavailable or congested, the content is automatically retrieved from another source.


Nevertheless, IPFS is not as technically complex as ICN. IPFS is open source software that works as a user application over the standard network services provided by any operating system. Compared to a full hardware and software ICN implementation, IPFS incurs in a modest performance hit in the host system and it is burdened by the IP protocol noticeable overhead. The experiment results of Section 7 demonstrate this effect. However, it has the advantage of being multi-platform (Windows, Linux, macOS, and Android) and compatible with existing IPv4 or IPv6 network infrastructure. IPFS software is still in development, but the codebase is stable and mature after years of uninterrupted testing on global public networks with thousands of live nodes across the world.

### 3.2. Inter-Planetary Name System (IPNS)

With IPFS, content is addressed by its hash, a mechanism that guarantees integrity but also limits content search. As the DHT-based search algorithm only takes hashes as input, nodes need a secure alternative for searching data in the network. The solution is IPNS: a system used to create and update mutable links. By using IPNS, each node can use its private and unique key to publish and sign data linked to its peer ID. This information is stored in the DHT and the node can update it as frequently as desired. As a result, any node that knows the peer ID of another node can make a request to the network and get the content of the latest information published by that node. Peer IDs are public and broadcasted to nearby nodes. The example described in Figure 5 shows how this system can be used to easily get data from a traffic camera:
The camera is an IPFS peer with a unique peer ID. The peer ID is a public hash generated from the private key of the camera. The private key is never shared with other peers.Every time that the camera captures an image, it generates its CID. CIDs are directly built using the content hash. The new CID is published and added to the distributed hash table (DHT).At this point, the user would be able to get the new camera image using the DHT; the problem is that the user does not know which is the CID of the image.To solve this problem, the camera regularly updates its IPNS address to point to the CID of the last captured image. This information is also published in the DHT and made available to all the nodes of the network.If the user makes a DHT request to resolve the camera IPNS address, it will always get the CID of the last camera image. The camera IPNS address is /ipns/<PeerID>. This IPNS address never changes and no other peer can update it because it would need to know the associated private key.The result of a DHT IPNS resolution is a CID, not the associated content. If the user wants to download the image, it can query the DHT for the list of peers that have a copy. The list will include the camera IP address and also other peers that already have retrieved the same image before.


### 3.3. RDF and JSON Linked Data

RSUs and vehicles use RDF to represent, share, and process semantic data. As mentioned before, RDF is not a syntax but an abstract language that W3C has standardized to support multiple syntax. The most common syntax is RDF/XML and IPFS would not have any problem dealing with this format. However, the IPFS developer community is very interested in semantic data and machine-to-machine interactions, and IPFS includes native support for RDF using the alternative JSON-LD (JSON Linked Data) syntax. Supported also by the W3C, JSON has several advantages over XML. For instance, it is easier to store and parse by computer programs. Besides, when a JSON document is shared by IPFS, IPFS does not treat the content as a plain text file but translates it first to the more efficient compressed version CBOR (Concise Binary Object Representation) [RFC 7049]. Compared to XML, the usage of CBOR can reduce network load up to a 65% [36].

When combined with IPNS, every peer would be able to publish a root JSON document with basic node information, and instead of creating a large root document, simply add links to additional documents with extended information. In IPFS, the links included in JSON documents are the CID of the referenced document. This allows nodes to add references to documents generated by any node in the network without the need of indicating a network address. Thanks to linked data, a semantic agent running in a vehicle or RSU would be able to efficiently navigate through the semantic data and easily find any kind of information published by any node in the network.

### 3.4. QUIC

Since version 0.5 (April 2020), IPFS has had support for using QUIC instead of Transmission Control Protocol (TCP) connections. QUIC is a transport-layer network protocol originally designed by Google, part of Chromium open-source code, and published as an Internet-Draft by the Internet Engineering Task Force (IETF) [37]. The QUIC protocol has been recently selected to replace TCP as the default HTTP (Hypertext Transfer Protocol) transport-layer protocol [38]. QUIC uses standard UDP datagrams and it is fully compatible with current IPv4 and IPv6 networks and equipment. Besides, the usage of UDP allows applications such as IPFS to implement QUIC without requiring an update of the underlying operating system. This is a summary of the advantages of QUIC over TCP connections:
Built-in security. QUIC connections provide TLS (Transport Layer Security) 1.3 authentication and encryption by default. QUIC is more efficient than TCP plus TLS. For example, a short HTTP request using TCP plus TLS requires two handshakes, for a total of eight messages. The same request with QUIC requires one handshake for a total of 5 messages.Improved multiplexing support. Applications can use a single QUIC connection to efficiently transmit multiple streams of data. QUIC keeps independent tracking of every stream, and when one frame is lost, only that frame is retransmitted and only one of the streams is affected. In general, QUIC has better loss recovery and congestion control than TCP.Connection migration. Every QUIC connections has an unique identifier. This makes possible to seamlessly continue an ongoing connection in the event of a network handover. This feature is especially useful in mobile networks (e.g., vehicular networks).Multipath. Although not yet supported, future QUIC implementations will also include the option to exchange data over multiple networks using a single connection [39]. This feature would increase the performance of IPFS in vehicular networks, as discussed in the experimental section of this paper.


## 4. Differences between a Static P2P Network and a Vehicular P2P Network

IPFS has been designed and it is being extensively used to create static P2P networks. However, this paper is considering the application of this P2P technology to vehicular networks. The main differences between a moving vehicle and a static computer are those related to the technologies used in the physical and data link layers. IPFS, notwithstanding, uses standard IP networking, which makes it largely independent from the lower layers. Any node that provides IPv4 or IPv6 connectivity, and supports TCP or QUIC connections, can use IPFS and its P2P services.

IPFS will have to use vehicle to anything (V2X) communications to interact with the rest of the vehicles and RSUs. Multiple network services are bound to coexist in the same vehicular network with different requirements in terms of priority, bandwidth, and latency. 5G will promote the communication capabilities to the next level, and provide a long-term roadmap for the enhancement of V2X communications. In the next deployment phase of ITS, environmental sensing will play a new role, in which all vehicles and the cloud share information about detected objects. This will enable vehicles to recognize obstacles hidden to their own sensors. This cooperative awareness will also permit semi-automated reactions, such as automatic braking for vulnerable road users’ (VRUs) protection and cooperative adaptive cruise control (C-ACC). These next phase services will require C-V2X connectivity, leveraging an associated Internet connection with a cloud service for cooperative knowledge sharing, and intelligent agents capable of collecting valid information, exchanging it with other peers, and making final decisions based on the processed data. One of the major improvements provided by 5G networks is network slicing [40], an advanced networking solution that allows the creation of flexible and differentiated services on top of a common network infrastructure.

Given that the communication between RSUs is expected to be more reliable and it permits higher throughput than vehicle to RSU connections, it seems reasonable to derive most of the traffic load to this static part of the network. With IPFS, this can be achieved by limiting the number of simultaneous connections established by vehicles, and prioritizing vehicle-to-RSU requests. Vehicles, especially while moving, would have to limit themselves to realize short content requests to nearby RSUs. Meanwhile, RSUs, fixed and capable of using high speed connections, would try to keep a large number of connections with other RSUs. Thanks to these connections, they would be able to detect changes in the data published by other RSUs, and get copies of all these data. The moment a vehicle establishes a connection with any RSU of the network, the vehicle would have immediate access to all the data published by nearby RSUs, without the need for establishing additional connections to other RSUs. In some rare cases, the vehicle could need to retrieve information that is not available in the already connected RSU. In this case, the IPFS distributed search algorithm returns the address of the nearest node that has a copy of the requested content. The vehicle has two options: establish a new connection with the indicated node, or use the already established connection as a relay, one of the advanced features of IPFS. By using a RSU as relay, the vehicle can get indirect access to all the nodes connected to that RSU.

In summary, the proposed semantic network is a hybrid network—part static, part mobile. The P2P network with IPFS would take advantage of the high speed and stability of the fixed connections between the multiple RSUs of the network. The RSUs would use these connections to share and distribute the semantic content. This part of the P2P network would have the properties and convenience of an static P2P network built using IPFS. The vehicles, however, have the limitations of a mobile and wireless connectivity. Thanks to the future 5G V2X networks, these vehicle-to-RSU connections are expected to have low bandwidth and excellent reliability, but IPFS can be configured to act differently in the moving vehicles. The vehicles would be able to connect to the network through any RSU, using it as a gateway to the rest of the P2P network, and participate only to get desired semantic information. In P2P terminology, the RSUs would act as seeders, providing all the information, and the vehicles as leechers, benefiting from their effort. This symbiosis is of particular relevance in the system proposed in this paper.

### Resource Allocation in Vehicular P2P Networks

Compared to traditional wired networks, resources in vehicular networks are very scarce. Providing satisfactory quality of service (QoS) becomes a challenging task due to the high mobility of the vehicles, and the limited spectrum. Therefore, it is essential to allocate and utilize the available wireless network resources in an appropriate manner. As comprehensively described in [41], multiple allocation methods are available or in development. With respect to the content-delivery service provided by the IPFS network, these would be the relevant requirements and constraints to consider when allocating network resources:
Vehicle-to-RSU communication. Satisfactory vehicle-to-RSU communication is the central pillar of the IPFS content-delivery network. RSU-to-RSU communication has higher bandwidth requirements, but it can be done outside the vehicular network. Vehicle-to-vehicle communication is not required for this service. The communication is solely unicast. Broadcast is not required.Latency. Low latency is important but not critical. The service does not have real-time requirements, and it should be considered less important than other services, such as emergency, safety, or platooning messages.Bandwidth. The content-delivery network is intended for compressed semantic data. As measured in Section 7, a typical transaction between vehicle and RSU involves just a few kilobytes of data.Packet loss and congestion. IPFS uses either TCP or QUIC connections. Both protocols provide retransmissions and advanced methods to avoid congestion. QUIC has been specially designed to cope with the limitations of mobile networks.Sudden network disconnections. Finally, IPFS is protected against abrupt disconnections. In those situations, the service would be temporarily suspended until the connection to the network is restored.


In summary, the content-delivery service provided by IPFS is extremely flexible, and it should be able to adapt to the allocated resources without a significant impact in the service. With unrestrained resources, the service is quick and responsive. With limited resources, the time required to retrieve content increases, but the process is never interrupted or compromised. Section 7 of this paper provides latency and bandwidth figures that could be used by network operators to select the appropriate resource allocation for this service.

## 5. Usage Scenarios

This section describes the particularities of several key scenarios found during the usage of the IPFS network built by the RSUs, and used by the vehicles. Figure 6 summarizes the four different usage scenarios.

### 5.1. Node Discovery

This first scenario introduces peer discovery, especially for vehicles. Discovery is the first step needed by all the nodes before connecting to a P2P network. IPFS already provides several alternatives for establishing the first connection to another node:
Bootstrap configuration peer list. The configuration file usually includes a list of peers with permanent and stable addresses. This list is used as a first connection attempt when a node is started.External discovery tool: Multicast DNS (mDNS) protocol. This IP-based discovery protocol can be used to detect other IPFS nodes in a local network. It does not work through NAT (network address translation), and therefore, it has a very limited reach capacity.Other external discovery tools using the IPFS application programming interface (API).


Once connected, the peers share information about other peers, which allows nodes to find any node of the network, despite its location. The bootstrap list is perfectly valid for RSUs due to the immobility of these nodes. RSUs should not have any issue connecting to a list of stable nodes and discovering the rest of the RSUs through these initial nodes. Although the vehicles could also use these initial lists to connect to the network, they can hugely benefit form the usage of external discovery tools.

Vehicles, initially disconnected from the IPFS network, have the opportunity to use a specific discovery protocol to find the peer IDs of the nearest RSUs. Although mDNS is an option, it is relatively slow and not designed for vehicular networks. Therefore, other protocols are desired. The most appropriate protocol would depend on the lower layers’ technologies. In case of using 4G or 5G V2X communications, vehicles could discover nearby IPFS nodes thanks to short-range sidelink communications [42]. Vehicles and RSUs would only need to include their IPFS addresses (a combination of their IP addresses and peer IDs) in the broadcast messages. These messages have a convenient short range that would avoid the unnecessary detection of IPFS nodes that are too far away to be useful.

### 5.2. RSU to RSU Distributed Sensor Data

In this usage scenario, multiple RSUs are interconnected and share their sensor data. RSUs establish connections based on content affinity. In other words, in a large RSU network formed by thousands of peers, every RSU has the possibility to connect to any of the other RSUs. Connecting to too many peers affects the performance and bandwidth usage, so it is necessary to limit the number of connections. IPFS automatically connects to peers when requesting content provided by them and disconnects when the peer is no longer useful. Therefore, RSUs would stay connected to those RSUs that have valuable content for them. That includes neighbor RSUs due to the interest in the data provided by nearby sensors, but also remote RSUs that are sited in key intersections or very transited roads.

While connected to a remote RSU, the local RSU immediately receives every content update published by the remote RSU. Connected is, therefore, the preferred state for neighbor RSUs. However, that does not mean that the local RSU cannot get content from other RSUs. The RSU can establish a new connection with any other RSU at any time and immediately get all the previously published content.

### 5.3. Vehicle to RSU Temporary Connection

In this scenario, one vehicle discovers one nearby RSU and requests data. Vehicles can connect to any node of the IPFS network. The point of entry is not important, and vehicles do not need to know the name or location of the data before connecting and doing a request. When the organization of the content is done using semantic structures and linked data, the relevant search parameters are not where the information is located but its meaning and context.

For example, a vehicle is approaching an intersection with the intention to turn right. Before arriving, the vehicle does an IPFS connection to get recent sensor information and decide the optimal lane and speed to use during the maneuver. If it is not connected to any peer yet, first it will need to discover the IPFS address of the nearest peers and connect to any of the discovered RSUs. Once connected, the next step is to get the RSU basic information. This information is a JSON document that should include at least the type of RSU, its location, its real-world address, and its capabilities. JSON is the syntax used to represent semantic data structured using RDF. The root document should never include a large quantity of information, but one or more links to additional documents instead. Vehicles can follow these links to have access to all the information published by the RSU. IPFS links are not URLs but hashes (CIDs). CIDs are unique for all the network and can be used to link to data published by any node in the network.

By default, RSUs should try to keep a local copy of all the linked documents, unless they do not have enough storage space. This problem could be quite common if the IPFS network is used to store large quantities of data, like video feeds produced by traffic cameras. In that case, an RSU linking to this data could decide to keep only copies of the most recent data. In case a vehicle requests access to data not available in the RSU, the DHT search would return the addresses of other nodes that still keep copies of the older data. Of course, these nodes do not need to be RSUs too, and could in fact be at any location. This opens the door to use remote cloud storage-dedicated nodes that would be only used to keep copies of old and rarely accessed data.

### 5.4. Vehicle to Vehicle Temporary Connection

Most of the time, IPFS would be used to fetch data collected and published by nearby RSUs. RSU’s published data could include, albeit not limited to, traffic information, weather conditions, video feeds, local events, and city details. However, any vehicle could also use IPFS to publish information and use local broadcasting to notify its IPFS address to nearby vehicles. Therefore, other vehicles could establish a connection and use it to get information. This information would also have to follow the same semantic organization used by RSUs, and therefore, vehicles would not have any problem understanding it. Due to privacy concerns, it is foreseeable that vehicles would limit themselves to share only anonymous data. Moreover, unlike RSUs, private vehicles do not have any interest in wasting network and storage resources keeping a copy of all the linked data. There are, however, other factors that could promote the usage of IPFS in vehicles. For example, local authorities could decide to use public vehicles, like buses, as active nodes, with the objective of reducing the load and number of active RSUs.

## 6. Security, Integrity, and Privacy

IPFS is designed to be a secure but open network. This section describes some of the security considerations that must be applied by all participants. The main characteristic of using a P2P network is that the data are decentralized and accessible to all peers. By using IPFS, data are guaranteed to not be deleted or modified, and there is no special node or backdoor with the ability to shut down the network or delete the data. Additionally, the usage of semantic data explicitly requires that all data are understandable by any of the agents; otherwise, some nodes would have partial or incomplete information. This requirement rules out the option of using encryption or adding support for different levels of access control, as suggested in [43]. In summary, the semantic IPFS network needs to be public and freely available for easy usage to all vehicles and RSUs. Otherwise, its purpose is unaccomplished. The principles of this service could be comparable to other public and freely available services, such as OpenStreetMaps.

Regardless of being designed as a public network, data integrity is desirable. IPFS guarantees data integrity because any content is addressed by its hash. If any node modifies the information before sending it, the receiver can detect the integrity failure by means of a quick hash function. If data source authenticity is needed, nodes could use a combination of IPNS and signed certificates. For instance, nodes could add to the content published under their IPNS addresses a link to a document listing all the previously authored content. Additionally, they could make available a certificate when a trusted certification authority (CA) signs the authenticity and veracity of the IPFS node. Note that IPNS addresses are unique, and only the original node owns the private key that allows publishing to its associated address.

Besides integrity, a primordial aspect of an open network is to provide privacy to its participants and avoid access to sensitive data. In the IPFS network, peers need to share their IP addresses and their peer IDs, whereas MAC addresses and other low-level information are not shared. IPFS does not require nodes to use a fixed public IP address, a common privacy concern, because the discovery of other IPFS nodes is done using local multicast or broadcast protocols. The IPFS peer ID is the cryptographic identity of a particular node and it is also used as IPNS address of the peer. The ID does not contain any private information and the node chooses what sensitive details it wants to publish under its IPNS address. However, the ID is unique and unconcealed to other peers, and therefore, it is feasible to track the node activity (published and requested data). Tracking RSU activity would be easy because there are always in the same location. However, tracking a vehicle becomes challenging in a P2P network because the vehicle is constantly changing its location and connecting to different peers. In order to become impossible to track, a vehicle should avoid sharing private information and it could decide to generate a new peer ID every time that it connects to and disconnects from the network.

## 7. Experimental Results

In the experiments, the Go implementation of IPFS has been used, specifically the September 2020 release 0.7. The initial test consisted of the capture of traffic generated by a real IPFS node over a span of 8 h. The scenario setup involved a default IPFS installation running over a Ubuntu 18.04 quad-core computer with 8 GB of RAM and a 100 Mbps fiber-optic Internet connection. During the test time, the node had complete freedom to create temporary connections and converse with the rest of the network. In the eight monitored hours, the local node exchanged data with 673 different peers around the world. However, 75% of the messages were exchanged with just 10 of those peers, with more than 50% of the total data being exchanged with just two particular peers. The inbound traffic was moderately low, while the outbound traffic was residual. This is an expected result given that the test node had to receive numerous updates from other nodes, but it did not publish any content. See Table 1 for additional results. The objective of this test was to measure the network usage of an idle node and to corroborate the stability of IPFS. Note that, during the realization of this test, the global network was still in the middle of the process to migrate to the QUIC protocol, and the majority of the nodes of the global network still preferred to use TCP connections.

For the rest of the experiments the setup was replaced by a private IPFS network, isolated from the global public network, formed by up to 30 nodes, each running a stand-alone Go-IPFS 0.7 client. The 30 nodes were built using Ubuntu 18.04 Docker containers. Each container was assigned a virtual IPv4, and connected to Docker virtual bridge. The containers setup and the virtual bridge were built over a single quad-core computer running Ubuntu 18.04 with 8 GB of RAM. The setup was limited to a maximum of 30 nodes to guarantee that the host computer did not run out of memory and processing power. In this setup, all the IPFS messages are exchanged in real-time. The performance of a Docker virtual network is only limited by the host CPU, with a measured end-to-end delay of less than 100 µs, and a measured throughput of more than 1 million packets per second. However, in order to simulate real-life V2X conditions, during the following tests, the end-to-end delay and the maximum throughput of the virtual bridge were programmatically altered using Linux Traffic Control capabilities.

For the first test with the private network, all the nodes were set to perform the same role. The scenario tries to emulate a private network formed by several RSUs. All the connections during the test were stable with no congestion and no message loss. After the initial network discovery, all the nodes got connected to each other, creating a fully connected mesh of n∗(n−1)/2 P2P connections (where *n* is the number of nodes). While all nodes were idle, the measurements showed that the total number of frames transmitted in the network was small, with less than one frame per second per active connection. IPFS default configuration is set to support large networks formed by thousands of nodes, and the default maximum number of simultaneous connections that a node allows is considerably higher than the 29 connections created during this test. The data exchanged while the 30 nodes were idle were residual and used to keep the network alive. The typical actions that could cause an spike in network traffic are the publishing and fetching of content. In the next step of the experiment, the impact of publishing content in the private network was tested. Each time a node adds new content, IPFS does not transmit the content itself to the other nodes. Instead, the network usage is caused by the update of the DHT. Every DHT update starts with several quick transactions between the origin node and each one of its connected peers. Immediately, this exchange is followed by additional transactions started by each one of the peers of the origin node, creating a controlled chain reaction. The effect of this chain reaction is shown in Figure 7. Most of the exchanged frames were very small, with an average packet size of 120 bytes.

### Discussion of Results

The comparison of results has been done against the mechanism proposed by X. Wang and Y. Li in [23] named CDVC (content delivery solution based on vehicular cloud) and the address-centric standard [RFC 7094] simulation measurements published in the same research. CDVC takes advantage of name-centric content search and delivery to reduce cost and latency. CDVC and the address-centric standard were simulated using the ns-2 network simulator. IPFS cost and latency values were measured using the real-time docker experiment described before. During the tests, the near-zero default docker end-to-end network delay was increased to match the values provided by the ns-2 simulation. All tests were performed using two different IPFS modes: TCP and QUIC.

In these tests, the role of one of the nodes was changed to be a vehicle. The end-to-end delay between the vehicle and the RSUs was configured to 5 ms in both directions. The first batch of tests was realized with the vehicle node initially disconnected from the IPFS network. The first task of the vehicle node was to connect to one of the RSUs. Immediately, it performed one or more content requests. Once the transaction was finished, the vehicle disconnected from the IPFS network. The results of these tests show that the cost of doing one single content request following this procedure with IPFS is higher than using CDVC or even the address-centric standard. However, the analysis of Figure 8 shows that the number of messages needed to retrieve data follows the equation:
(1)$$m=m_{0}+r*m_{r}$$
where the total number of messages (*m*) is equal to the fixed amount of messages ($m_{0}$) needed to establish a connection to the IPFS network, plus the variable cost of doing each additional content requests (*r*). The value of $m_{r}$ is considerably lower when IPFS is working on QUIC mode, 12 messages per request, versus the 25 messages per request needed when using TCP mode. Therefore, there is a huge improvement using QUIC. However, the fixed cost of the connection is higher with QUIC, 60 messages, versus 40 using TCP.

During the tests, the RSU nodes were configured to generate content with an average size of 1000 bytes. The content size does not have any impact in the number of exchanged messages, as long as the maximum message size is lower than the network’s maximum transmission unit (MTU), set to the Ethernet standard 1500 bytes during our tests. Otherwise, large messages would need to be fragmented by the network layer. The data format used, CBOR, provides a high compression rate for semantic data,, and therefore, RSUs would rarely need to transmit large content messages to the vehicles. However, the content size does have a direct impact on the number of transmitted bytes. Considering an average size of 1000 bytes per content request, Figure 9 shows that TCP messages had an average size of 109 bytes, while QUIC messages had an average size of 125 bytes. Those numbers actually show that QUIC is more efficient. TCP’s average size is smaller because it requires more small protocol-related messages than QUIC. Figure 10 shows that, despite the reduced number of messages and bytes, IPFS needed approximately the same time to process the *n* content requests for both connection modes. Without taking into account the time needed to connect and disconnect from the network, with an estimated value of 200 ms, every additional content request needed an average of 20 ms. The reason for this contradictory result is that IPFS is CPU bound: the latency is mostly caused by the time needed by the CPU to process and generate the content. Besides, QUIC is a new transport layer protocol and its CPU usage is not yet as optimized as TCP for Linux operating systems.

When comparing IPFS experiment results with the results obtained with the simulations, it is important to take into account that these simulations did not include any kind of encryption nor authentication procedure. The access-centric standard simulation performed the request using a plain and insecure TCP connection. CDVC, however, used the Content-Centric Networking (CCN) protocol stack proposed by V. Jabobson et al. at [44]. Therefore, CDVC did not include the authentication of the participant nodes, the vehicle did not require any kind of initial connection to the network, and the request did not have the overhead of a TCP or QUIC connection. The fact that IPFS uses TLS for encryption and authentication, and the connection-oriented nature of this protocol, make IPFS less efficient when only one small content request is done by the vehicle. The Table 2 shows a comparison between IPFS, CDVC, and the standard address-centric protocol. IPFS is shown in two rows, one with the cost of doing a single and isolated content request, and the second when multiple requests are done. The results show that the real implementation of IPFS is capable of matching, during the operation of the system, the best performance shown in the state-of-the-art, even when compared with a simulation, which includes many simplifications that clearly make it compete in better conditions.

## 8. Conclusions

In this paper, we have proposed the creation of a P2P content-delivery network for vehicles and RSUs. The content of this network would be organized as semantic data, with all their associated advantages and benefits. In order to create this network, we have selected the open-source IPFS protocol. IPFS networks provide P2P search and data distribution over standard IP networks. However, the protocol has not been designed with vehicular networks in mind. The aim of this paper has been, hence, to study the possibility of using IPFS for the proposed vehicular network service.

The current research and experimental results have shown that it is easy to use IPFS to create and validate a content-addressed network formed by up to 30 peers. It is also possible to test it against the public IPFS network, formed by thousands of peers. IPFS uses standard IP technology to provide content-based addressing in opposition to host-based solutions. This feature would allow vehicles, RSUs, and other IoT devices to directly share content with any other node, without needing to connect to any server, or know the IP addresses or domain names of their neighbors. This paper has also shown that the resultant P2P network offers most of the functionality advantages of the highly regarded information-centric networking technologies.

Vehicles would be able to connect to the IPFS network through any RSU and easily fetch the semantic content shared by any node of the network. Three key and unique features make IPFS especially suitable for this application compared to other P2P solutions. First, its native and efficient support of semantic data using JSON. Second, the concept of IPNS address: every node has an exclusive address to publish secure and trusted content, reachable by anyone, without the need for centralized servers. Third, the recent addition of the QUIC protocol, more efficient and robust than the traditional use of TCP or UDP connections.

The experimental results have been compared with recent simulations of both content-based and host-based algorithms. The analysis of the results has revealed that IPFS incurs a noticeable penalty during the initial connection, mostly due to the usage of TLS for authentication and encryption. These are two security features that were not included in the state-of-the-art simulations used for comparison. When IPFS is used to get multiple content requests, IPFS cost quickly decreases, reaching the excellent numbers estimated by the CDVC simulation. IPFS is inefficient when a vehicle needs to retrieve just one message with a few bytes of information, but highly cost-effective when the data required by the vehicle are fragmented in tenths of small JSON files. It is important to stress that those are exactly the kind of requests that a semantic agent running on the vehicle would do. In summary, it would be inefficient for vehicles to use IPFS to get a single content request, but that is going to be extremely rare in a semantic case. Semantic data are formed by tenths of interlinked entries. A vehicle requesting data would start doing a single request, process it, and immediately perform multiple requests to get all the related data, without disconnecting from the network. IPFS is perfectly suited for requests of this kind, as it provides immediate access to the requested data, and optimizes the network usage when multiple simultaneous requests are performed.

The main conclusion of this paper is that IPFS is a very promising solution for exchanging semantic data in vehicular networks. The resultant P2P network offers a great balance between functionality, performance, and security. The results have been limited to a few specific situations in order to be able to compare them with state-of-the-art simulations. However, future lines of research should extend these experiments to more complex situations, including real semantic data requests, and evaluating the possible shortcomings of having multiple vehicles performing simultaneous connections, with the goal of analyzing the real-time performance of a full-fledged vehicular network. These new experiments should make use of future IPFS software releases and newer Linux versions. Performance improvements are expected in both areas, now that QUIC is becoming the default protocol in all Web-related technologies.

## Figures and Tables

**Figure 1 sensors-20-06404-f001:**
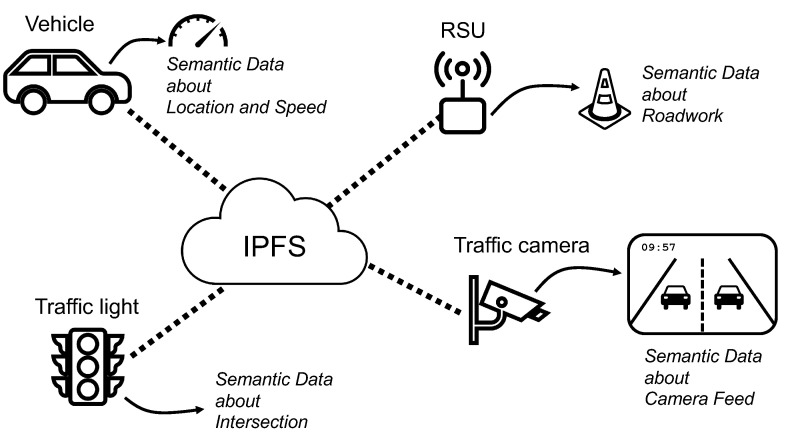
Vehicles, road-side units (RSUs), and other IoT devices (e.g., traffic cameras and traffic lights) connected to the peer-to-peer (P2P) Inter-Planetary File System (IPFS) network to share and retrieve semantic data.

**Figure 2 sensors-20-06404-f002:**
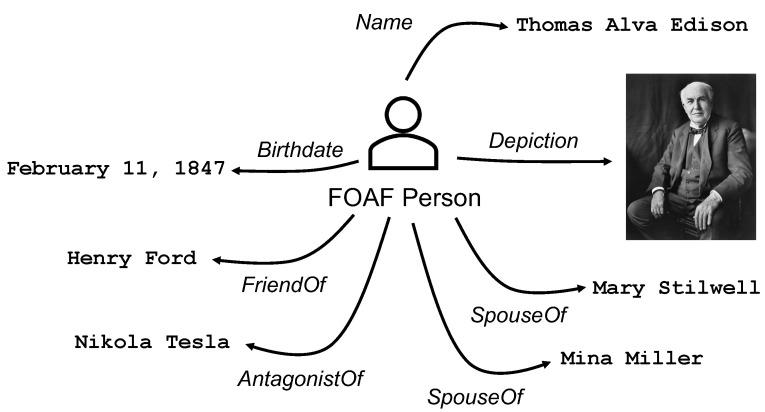
This example shows how to use semantic data and the Friend of a Friend (FOAF) ontology to describe people and their relationships. Image of Thomas Edison from the Library of Congress, copyright by Bachrach (circa 1922).

**Figure 3 sensors-20-06404-f003:**
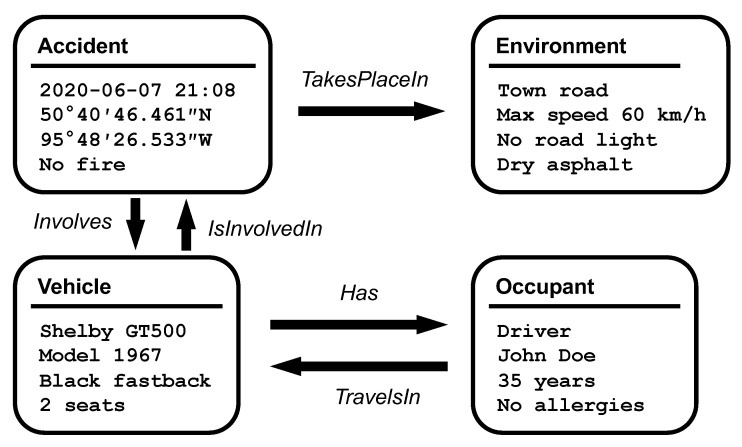
This example uses a subset of the Car Accident Ontology for VANETs (CAOVA) ontology to describe a car accident.

**Figure 4 sensors-20-06404-f004:**
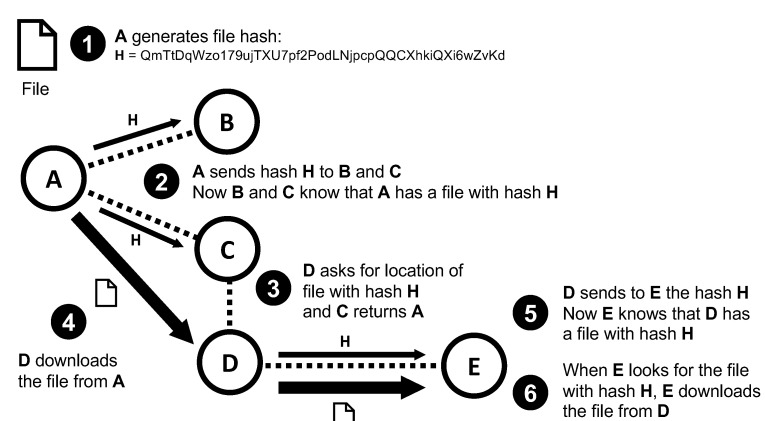
The peer A is the creator of the file. After sharing the hash, D can download the file from A. Immediately, D notifies E that it can download the file from it without connecting to A.

**Figure 5 sensors-20-06404-f005:**
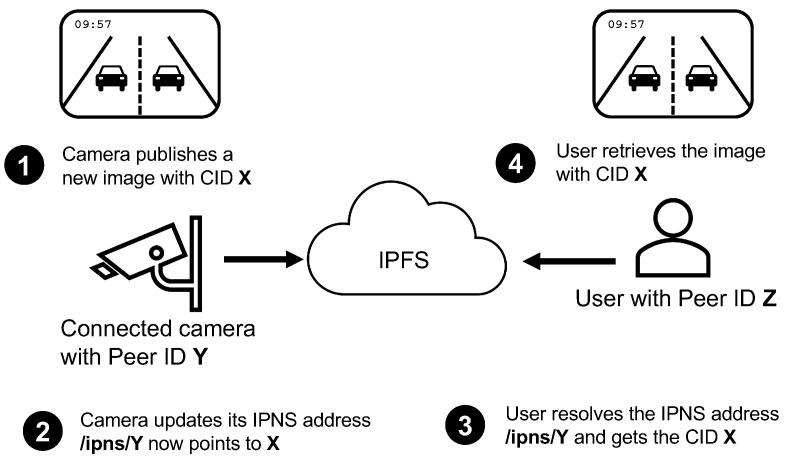
A camera publishes a new image using its peer ID and the user resolves the camera Inter-Planetary Name System (IPNS) address to fetch the latest published image.

**Figure 6 sensors-20-06404-f006:**
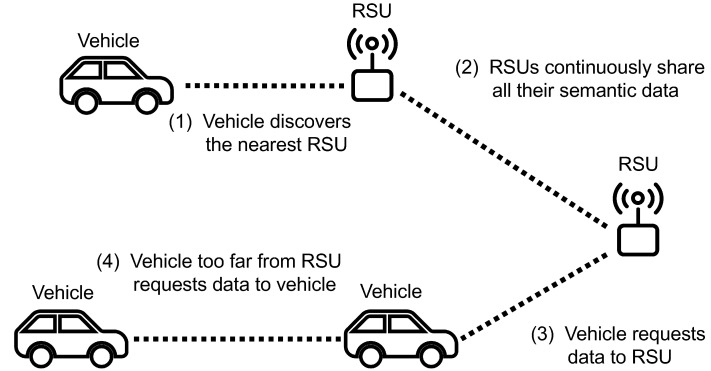
Graphic summary of the four different usage scenarios.

**Figure 7 sensors-20-06404-f007:**
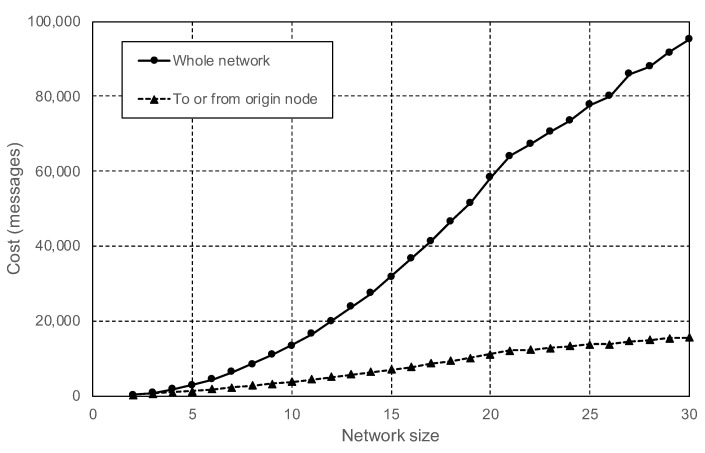
Total of messages exchanged in the peer network when content is added.

**Figure 8 sensors-20-06404-f008:**
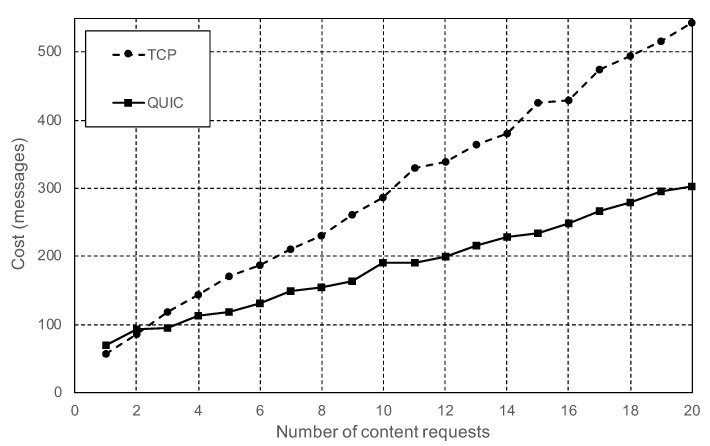
Total number of messages required to fetch content from another node, including the initial connection.

**Figure 9 sensors-20-06404-f009:**
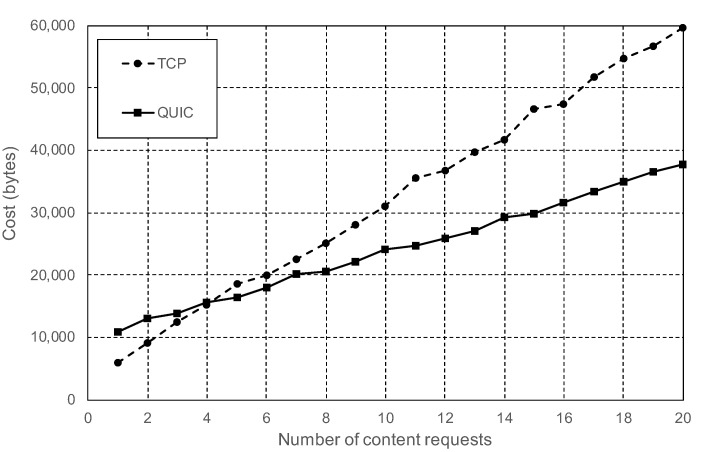
Total number of bytes required to fetch content from another node including the initial connection.

**Figure 10 sensors-20-06404-f010:**
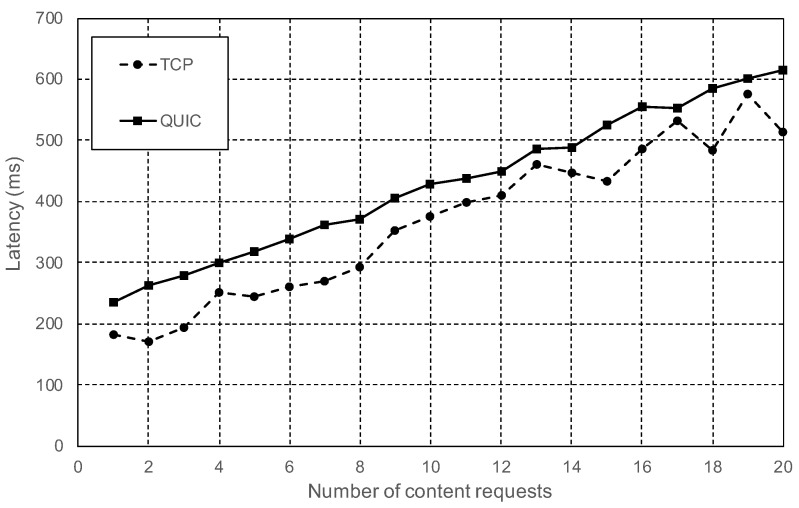
Total time required to fetch content from another node, including the initial connection.

**Table 1 sensors-20-06404-t001:** Results of running an idle IPFS node for 8 h.

Measurement	Transmission	Reception
Total frames	573,312 frames	552,750 frames
Average frame size	77 bytes	153 bytes
Average frames per second	19.0 fps	19.2 fps
Average throughput	12 kbps	23 kbps
Maximum throughput peak	462 kbps	2.8 Mbps

**Table 2 sensors-20-06404-t002:** Content delivery comparison.

Method	Cost (Messages)	Latency (ms)
Address-Centric	27	75
CDVC	13	30
IPFS TCP (first)	57	183
IPFS QUIC (first)	69	235
IPFS TCP (average)	27	28
IPFS QUIC (average)	15	30

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
