# Peer review of "Semantic Distributed Data for Vehicular Networks Using the Inter-Planetary File System"

_sensors, 2020, doi:10.3390/s20226404_

Round 1
Reviewer 1 Report
The paper proposes the semantic P2P vehicular network using IPFS and QUIC. The authors have updated and extended their paper according to my previous review comments.
Author Response
Dear Reviewer, thank you very much for reading and reviewing our manuscript. We are very glad to have finally met your expectations. Thanks again.
Reviewer 2 Report
Reviews of the work are nice so far. I approve on publication
Author Response
Dear Reviewer, thank you very much for reading and reviewing our manuscript. We are very glad to have your approval. Thanks again.
Reviewer 3 Report
The paper presents the creation of a semantic distributed network using content-addressed networking and peer-to-peer (P2P) connections. In this paper, Road-Side Units (RSUs) and vehicles exploited ontologies to semantically represent information and facilitate the development of intelligent autonomous agents capable of navigating and processing the shared data. The Authors used the interplanetary file system (IPFS) to provide secure, reliable, and efficient content-addressed distributed storage over standard IP networks using the new QUIC protocol. According to Authors, the results of the experiments confirmed that IPFS is a promising technology that provides a perfect balance between functionality, performance and security. The topic is interesting and the paper is well corresponding to the journal aim and scope.
The paper is well structured. Part of the article is a theoretical study, the practical part begins with section 5 and includes usage scenarios. In this article, the Authors described in details the used components and provided the motivation to use a specific component in their solution.
However there are shortcomings in this paper. The article also lacks a graphic visualization of the procedure - it would certainly facilitate understanding and improve readability. In conclusions section please also mention the aim of the work.
For the purpose of verification, the feasibility of this proposal was checked and compared with the simulation results published by X.Wang and Y. Li. Is it possible to make a comparison for the other simulation results? It would be useful to consider other approaches to assess the benefits. An interesting added value would be to extend the experiments on a larger scale with large amounts of semantic data and additional nodes - the Authors briefly mentioned this in the summary.
Author Response
Dear Reviewer, thank you very much for reading and reviewing our manuscript. We appreciate your kind words. Please find below our reply to your comments:
Reviewer's Comment
However there are shortcomings in this paper. The article also lacks a graphic visualization of the procedure - it would certainly facilitate understanding and improve readability. In conclusions section please also mention the aim of the work.
Authors' Reply
Following the Reviewer request, we have created and added a new Figure 6 (line 406) that graphically summarizes the four described scenarios in section 5. The text of the section has been also adapted accordingly to improve the readability and understandability of the paper.
In order to mention the aim of the work, the following paragraph have been added to the conclusions (line 619):
“In this paper, we have proposed the creation of a P2P content-delivery network for vehicles and RSUs. The content of this network would be organized as semantic data, with all their associated advantages and benefits. In order to create this network, we have selected the open-source IPFS protocol. IPFS networks provide P2P search and data distribution over standard IP networks. However, the protocol has not been designed with vehicular networks in mind. The aim of this paper has been hence to study the possibility of using IPFS for the proposed vehicular network service.”
Reviewer's Comment
For the purpose of verification, the feasibility of this proposal was checked and compared with the simulation results published by X.Wang and Y. Li. Is it possible to make a comparison for the other simulation results? It would be useful to consider other approaches to assess the benefits. An interesting added value would be to extend the experiments on a larger scale with large amounts of semantic data and additional nodes - the Authors briefly mentioned this in the summary.
Authors' Reply
We apologize for the limited comparison of the results of our experiments with other related research results. Due to the nature of our research, the results are difficult to compare with the results presented by previous publications. The main reason is that, in these experiments, we are not getting simulation results but the results of capturing the performance of a real-time experiment with a fully functional software. That is, unfortunately, a fairly uncommon practice in this field of research (P2P content-delivery) because other authors are using in-development content-delivery protocols that are only possible to test using simulations, and, when using simulations, it is very impractical to test large networks and large quantities of data. However, by using IPFS, we are unable to do a network simulation, but we have the possibility to test multiple quasi-real nodes thanks to Docker virtualization. We have focused the main part of our experiments in trying to emulate the simulation scenario proposed by X.Wang and Y. Li, but, if we limit our results to this comparison, the section would only include a quite narrow experiment, focused in a small particular situation that does not reflect the reality and complexity of a content-delivery P2P network. For that reason, we have tried to expand the scope of the experiments, with relevant additional results, easy to understand, but with results that cannot be compared with any other existent research because, as far as we know, there are no equivalent experiments. We know it is difficult for the reader to assess the advantages of IPFS without a direct comparison, but we hope that, in the near future, more researchers would start to use IPFS or similar P2P solutions to provide and expand the scope of this kind of experiment results, and our publication would then be useful for comparison. And, in the near future, when more researchers publish different solutions and results, we will be glad to prepare a new paper with additional improvements and comparisons. The section of conclusions has also included this rationale.
As the reviewer kindly points, we briefly suggest the extension of the experiments to a larger scale. Unfortunately we have not been able to perform them yet due to time and resource constraints. Besides, we would face the same dilemma, we would have some interesting results, but we would not be able to compare them with any other research. However, if this paper is published, we plan to continue this line of research and provide additional scenarios, procedures, and large scale results.
Thanks again.
Reviewer 4 Report
A good-quality scientific article, however, raises two comments:
- Not all abbreviations used in the text are included in the "Abbreviations" section. All abbreviations are either explained in the text or self-explanatory and in common use. Perhaps it is better to give up this section altogether.
- The question is whether the "References" section of 60 items is really needed for this article. It is worth considering limiting it, which will contribute to greater precision and readability of the text.
- The graphic visualization of the presented procedure is slightly lacking, such visualization would improve readability.
- The Conclusions seem somewhat laconic and do not fully define the value of the results obtained. It is also worthwhile to briefly indicate the purpose and scope here.
Author Response
Dear Reviewer, thank you very much for reading and reviewing our manuscript. Please find below our reply to your comments:
Reviewer's Comment
- Not all abbreviations used in the text are included in the "Abbreviations" section. All abbreviations are either explained in the text or self-explanatory and in common use. Perhaps it is better to give up this section altogether.
Authors' Reply
Thanks for pointing the issue. IPFS acronym now is defined in the line 56. It is also defined in the abstract, but this is the first time that appears in the article text.
ETSI ITS-G5 and 3GPP C-V2X acronyms are now defined in the line 67.
Following the reviewer’s suggestion the “Abbreviations” section has been completely removed. Please note that, although QUIC was initially proposed as the acronym for "Quick UDP Internet Connections", now it is not considered an acronym by the IETF, it is simply the name of the protocol.
Reviewer's Comment
- The question is whether the "References" section of 60 items is really needed for this article. It is worth considering limiting it, which will contribute to greater precision and readability of the text.
Authors' Reply
We completely agree with the Reviewer comment. The usage of 60 referenced items was uncalled and confusing for the reader. Due to manuscript changes requested by another reviewer, a new reference has been added. However, we have been able to remove 15 references. Some of them have been replaced with a short mention to the W3C recommendation name or RFC number.
This is the complete list of removed references and the reason to remove it:
This reference has been removed because it is out of the scope of this paper to provide a comparison between V2X communication systems:
Mannoni, V.; Berg, V.; Sesia, S.; Perraud, E. “A Comparison of the V2X Communication Systems:ITS-G5 and C-V2X”. 2019 IEEE 89th Vehicular Technology Conference (VTC2019-Spring), 2019, pp. 1–5.
This reference has been removed because other references already include enough information about blockchain IoT applications:
Ferrag, M.A.; Derdour, M.; Mukherjee, M.; Derhab, A.; Maglaras, L.; Janicke, H. “Blockchain Technologies for the Internet of Things: Research Issues and Challenges”. IEEE Internet of Things Journal 2019,6, 2188–2204.
This reference has been removed because it is not an article but just the name of the FOAF specification. The FOAF specification is easy to find in Internet just looking for FOAF:
Brickley, D.; Miller, L. The Friend Of A Friend (FOAF) Vocabulary Specification, 2014.
These 4 references have been removed because they are not articles but W3C recommendations. The name of the recommendation name is enough for finding the full text in Internet:
Semantic Sensor Network Ontology W3C recommendation, W3C, 2017.
OWL 2 Web Ontology Language Document Overview (Second Edition). W3C recommendation,682W3C, 2012.
RDF 1.1 XML Syntax. W3C recommendation.
JSON-LD 1.1. W3C candidate recommendation.
This reference has been removed because, although it is useful as introduction to XML and RDF, it is not relevant for this research:
Decker, S.; Melnik, S.; van Harmelen, F.; Fensel, D.; Klein, M.; Broekstra, J.; Erdmann, M.; Horrocks, I. “The Semantic Web: the roles of XML and RDF”. IEEE Internet Computing 2000,4, 63–73.
These three references are removed because they were included to give some background to the reader, but there are very well-known applications and it is very easy to find information just looking in Internet:
Pouwelse, J.; Garbacki, P.; Epema, D.; Sips, H. “The Bittorrent P2P File-Sharing System: Measurements and Analysis”. Peer-to-Peer Systems IV 2005, 3640, 205–216.
Thung, F.; Bissyandé, T.; Lo, D.; Jiang, L. “Network Structure of Social Coding in GitHub”. Proceedings of the Euromicro Conference on Software Maintenance and Reengineering, CSMR2013, pp.323–326.
Haklay, M.; Weber, P. “OpenStreetMap: User-Generated Street Maps”. IEEE Pervasive Computing 2008,7177, 12–18.
This reference has been removed because is redundant:
Shi, W.; Cao, J.; Zhang, Q.; Li, Y.; Xu, L. “Edge Computing: Vision and Challenges”. IEEE Internet of Things Journal 2016,3, 637–646.
These references have been replaced by the RFC number:
Bormann, C.; Hoffman, P. Concise Binary Object Representation (CBOR). RFC 7049, IETF, 2013.
McPherson, D.; Oran, D.; Thaler, D.; Osterweil, E. Architectural Considerations of IP Anycast. RFC 7094, IETF, 2014.
Reviewer's Comment
- The graphic visualization of the presented procedure is slightly lacking, such visualization would improve readability.
Authors' Reply
Following the Reviewer's request, we have created and added a new Figure 6 (line 406) that graphically summarizes the four described scenarios in section 5. The text has been updated accordingly to improve the readability of the paper.
Reviewer's Comment
- The Conclusions seem somewhat laconic and do not fully define the value of the results obtained. It is also worthwhile to briefly indicate the purpose and scope here.
Authors' Reply
In order to mention the purpose and scope of the work, the following paragraph have been added to the conclusions (line 619):
“In this paper, we have proposed the creation of a P2P content-delivery network for vehicles and RSUs. The content of this network would be organized as semantic data, with all their associated advantages and benefits. In order to create this network, we have selected the open-source IPFS protocol. IPFS networks provide P2P search and data distribution over standard IP networks. However, the protocol has not been designed with vehicular networks in mind. The aim of this paper has been hence to study the possibility of using IPFS for the proposed vehicular network service.”
The paragraph starting at line 656 has been updated to mention the major limitation of this study and remark possible future extensions:
“The results have been limited to a few specific situations in order to be able to compare with state of the art simulations. However, future lines of research should extend these experiments to more complex situations, including real semantic data requests, and evaluating the possible shortcomings of having multiple vehicles performing simultaneous connections, with the goal of analyzing the real-time performance of a full-fledged vehicular network.”
Following the Reviewer suggestion, in order to give better understanding of the results to the reader, and remark the advantages of IPFS over the protocols used by the simulations, we have added this text to the paragraph starting at the line 641:
“[...] In summary, it would be inefficient for vehicles to use IPFS to get a single content request, but that is going to be extremely rare in a semantic case. Semantic data are formed by tenths of interlinked entries. A vehicle requesting data would start doing a single request, process it, and immediately perform multiple requests to get all the related data, without disconnecting from the network. IPFS is perfectly suited for this kind of requests, as it provides immediate access to the requested data, and optimizes the network usage when multiple simultaneous requests are performed.”
Thanks again.
Reviewer 5 Report
Article is interesting, the subject is current and has useful value. The citations and references are valid and relevant to the text and bibliographic research and references appear complete and satisfactory. In order to enhance the article quality, I suggest the following remarks be taken into account:
- Please highlight the elements of the novelty.
- Please give the full name of the acronyms the first time they appear in the text (for instance IPFS).
- Figure 2 seems to be insufficiently described.
- Why is assumed 30 peers in the experiments? Please explain.
- The authors should add equation in line 541.
Author Response
Dear Reviewer, thank you very much for reading and reviewing our manuscript. We appreciate your kind words. Please find below our reply to your comments:
Reviewer's Comment
Please highlight the elements of the novelty.
Authors' Reply
In order to mention the aim and novelty of the work, the following paragraph have been added to the conclusions (line 619):
“In this paper, we have proposed the creation of a P2P content-delivery network for vehicles and RSUs. The content of this network would be organized as semantic data, with all their associated advantages and benefits. In order to create this network, we have selected the open-source IPFS protocol. IPFS networks provide P2P search and data distribution over standard IP networks. However, the protocol has not been designed with vehicular networks in mind. The aim of this paper has been hence to study the possibility of using IPFS for the proposed vehicular network service.”
Reviewer's Comment
Please give the full name of the acronyms the first time they appear in the text (for instance IPFS).
Authors' Reply
IPFS acronym now is defined in the line 56. It is also defined in the abstract, but this is the first time that appears in the article text.
ETSI ITS-G5 and 3GPP C-V2X acronyms are now defined in the line 67.
Please note that although QUIC was initially proposed by Google as the acronym for "Quick UDP Internet Connections", it is not considered an acronym by the IETF, and it is simply the name of the protocol.
Reviewer's Comment
Figure 2 seems to be insufficiently described.
Authors' Reply
The text in the line 118 has been extended to better describe the Figure 2:
“The Figure 2 shows a basic usage of semantic data describing a person using the Friend Of A Friend (FOAF) ontology. Semantic data is represented by triples (subject-predicate-object). In this example, the subject is the depicted person, the predicate is indicated with an arrow, and the object is the pointed value. For example, the person's name is Thomas Alva Edison and this person is the friend of Henry Ford. This generic structure can be used to describe highly complex concepts and their relationships”
Reviewer's Comment
Why is assumed 30 peers in the experiments? Please explain.
Authors' Reply
We have added this remark in the line 536: “The setup was limited to a maximum of 30 nodes to guarantee that the host computer did not run out of memory and processing power.”. That is actually the reason to limit the tests to 30 nodes. In order to have more nodes, it would have been necessary to prepare more CPU and RAM memory. That is really not difficult, the test uses just a quad-core processor and 8 GB of RAM. But the results show that it was not meaningful to add more nodes to the test.
Reviewer's Comment
The authors should add equation in line 541.
Authors' Reply
Thanks for pointing out this. The wrongly formatted equation of the line 541 (now line 579 after other changes) has been rewritten and the related text has been updated to further clarify the equation's meaning.
Thanks again.
Reviewer 6 Report
This paper studies the creation of a semantic distributed network using content-addressed networking and peer-to-peer (P2P) connections. I have few general comments that need to be addressed carefully before I can recommend for publications:
(1) What is the impact of file size in the proposed scheme?
(2) For which V2X technology the proposed technique is applicable? CV2X or DSRC?
(3) Resources in the V2X network is very limited. A discussion is required on the choice of appropriate Resource allocation technique based on following papers:
(i) Noor, "A Survey on Resource Allocation in Vehicular Networks," IEEE Transactions on Intelligent Transportation Systems, 2020.
(ii) Rui, "Distributed resource allocation for D2D communication underlaying cellular networks," 2013 IEEE International Conference on Communications Workshops (ICC).
(4) Please mention the limitations, challenges, and possible future extensions.
Author Response
Dear Reviewer, thank you very much for reading and reviewing our manuscript. Please find below our reply to your comments:
Reviewer's Comment
(1) What is the impact of file size in the proposed scheme?
Authors' Reply
We apologize for not including this information in the original manuscript. We have extended the paragraph staring at line 587 with additional details:
“During these tests, the RSU nodes have been configured to generate content with an average size of 1000 bytes. The content size does not have any impact in the number of exchanged messages, as long as the maximum message size is lower than the network Maximum Transmission Unit (MTU), set to the Ethernet standard 1500 bytes during our tests. Otherwise, large messages would need to be fragmented by the network layer. The data format used, CBOR, provides a high compression rate for semantic data, and, therefore, RSUs would rarely need to transmit large content messages to the vehicles. However, the content size does have a direct impact on the number of transmitted bytes. Considering an average size of 1000 bytes per content request, the Figure 8 shows that TCP messages had an average size of 109 bytes, while QUIC messages had an average size of 125 bytes. This number actually shows that QUIC is more efficient. TCP average size is smaller because it requires more small protocol-related messages than QUIC. The Figure 9 shows that, despite the reduced number of messages and bytes, IPFS needed approximately the same time to process the n content requests for both connection modes. Without taking into account the time needed to connect and disconnect from the network, with an estimated value of 200 ms, every additional content request needed an average of 20 ms. The reason for this contradictory result is that IPFS is CPU bound: the latency is mostly caused by the time needed by the CPU to process and generate the content. Besides, QUIC is a new transport layer protocol and its CPU usage is yet not as optimized as TCP by Linux operating systems.”
In summary, the size does not have a relevant impact on the number of messages and the total latency. We do not have exact numbers, but semantic data is mostly compressed text and it is usually formed by small messages (a few hundred bytes as maximum). We have considered it more relevant to show results with multiple content requests than different content sizes because a typical transaction would probably need several contiguous but independent content requests.
Reviewer's Comment
(2) For which V2X technology the proposed technique is applicable? CV2X or DSRC?
Authors' Reply
Anyone. IPFS is an application layer protocol that uses standard IP sockets (this is mentioned at the line 65). It can work using TCP connections and UDP connections (QUIC protocol is actually layered over UDP sockets). Any network technology that supports TCP or UDP connections can support IPFS. That includes CV2X and DSRC.
We have included DSRC in the list of IPFS-compatible lower layer protocols mentioned in the paragraph starting at the line 65:
“IPFS is a multi-platform software that works over standard IP networks. Therefore, it allows RSUs and other IoT devices to interact independently of their connection to the network. Vehicles can use specific wireless systems, such as Dedicated Short-Range Communications (DSRC), the European Telecommunications Standards Institute (ETSI) Intelligent Transport Systems (ITS) G5 standard, or the 3rd Generation Partnership Project (3GPP) Cellular vehicle-to-everything standard, while RSUs can make use of multiple interfaces, wireless and wired, to connect simultaneously to vehicles, other RSUs, and IoT devices as traffic cameras.”
Reviewer's Comment
(3) Resources in the V2X network is very limited. A discussion is required on the choice of appropriate Resource allocation technique based on following papers:
(i) Noor, "A Survey on Resource Allocation in Vehicular Networks," IEEE Transactions on Intelligent Transportation Systems, 2020.
(ii) Rui, "Distributed resource allocation for D2D communication underlaying cellular networks," 2013 IEEE International Conference on Communications Workshops (ICC).
Authors' Reply
Thanks for suggesting these papers. Thanks to them, we have been able to add a new meaningful subsection to the paper. The new subsection starts at the line 372:
“Resource Allocation in Vehicular P2P Networks
Compared to traditional wired networks, resources in vehicular networks are very scarce. Providing satisfactory Quality of Service (QoS) becomes a challenging task due to the high mobility of the vehicles, and the limited spectrum. Therefore, it is essential to allocate and utilize the available wireless network resources in an appropriate manner. As comprehensively described in [43], multiple allocation methods are available or in development. With respect to the content-delivery service provided by the IPFS network, these would be the relevant requirements and constraints to consider when allocating network resources:
- Vehicle-to-RSU communication. A satisfactory vehicle-to-RSU communication is the center pillar of the IPFS content-delivery network. RSU-to-RSU communication has higher bandwidth requirements, but it can be done outside of the vehicular network. Vehicle-to-vehicle communication is not required for this service. The communication is solely unicast. Broadcast is not required.
- Latency. Low latency is important but not critical. The service does not have real-time requirements, and it should be considered less important than other services as emergency, safety, or platooning messages.
- Bandwidth. The content-delivery network is intended for compressed semantic data. As measured in section \ref{Results}, a typical transaction between vehicle and RSU involves just a few kilobytes of data.
- Packet loss and congestion. IPFS uses TCP or QUIC connections. Both protocols provide retransmissions and advanced methods to avoid congestion. QUIC has been specially designed to cope with the limitations of mobile networks.
- Sudden network disconnections. Finally, IPFS is protected against abrupt disconnections. In those situations, the service would be temporarily suspended until the connection to the network is restored.
In summary, the content-delivery service provided by IPFS is extremely flexible, and it should be able to adapt to the allocated resources without a significant impact in the service. With unrestrained resources, the service is quick and responsive. With limited resources, the time required to retrieve content is increased, but never interrupted or compromised. The section 7 of this paper provides latency and bandwidth numbers that could be used by network operators to select the appropriate resource allocation for this service.”
Reviewer's Comment
(4) Please mention the limitations, challenges, and possible future extensions.
Authors' Reply
The paragraph starting at line 656 has been updated to mention the major limitation of this study and remark possible future extensions:
“The results have been limited to a few specific situations in order to be able to compare with state of the art simulations. However, future lines of research should extend these experiments to more complex situations, including real semantic data requests, and evaluating the possible shortcomings of having multiple vehicles performing simultaneous connections, with the goal of analyzing the real-time performance of a full-fledged vehicular network.”
One of the major challenges found during this research have been finding a common ground to compare with state of the art simulations. This is an original research based on a real software protocol never used before in this field of research. For that reason, every experiment had to be designed to not deviate in excess from the typical vehicular network simulation. Otherwise, the results would have become unmeaningful to readers as they would not be able to assess them.
The other major challenge is related also to the experiment. It is always more difficult to obtain meaningful and repeatable results from a real software and a real (although virtualized) setup than a closed and clearly constrained simulation. We have needed to automate all the experiments in order to be able to adjust and repeat them hundreds of times until the results have been precise and consistent.
We consider that it is not adequate to mention these challenges in the paper because we have been able to overcome them and they do not affect the results nor the conclusions.
Thanks again.
Round 2
Reviewer 6 Report
I am satisfied with the current version.
This manuscript is a resubmission of an earlier submission. The following is a list of the peer review reports and author responses from that submission.
Round 1
Reviewer 1 Report
First of all the main contribution of this paper should be outlined. It is not clear what is the main goal of this work. It seems that the paper duscusses only the application of IFPS to Vehicle Networks but it does not present any new ideas or solutions.
Author Response
Dear Assistant Editor and Reviewers,
We would like to thank the Assistant Editor and the Reviewers once again for helping us to see our work from the reader's perspective and to improve its exposition accordingly. We have modified the manuscript based on the provided comments, and we hope that the modifications that we have made and the responses that we have provided herein will address the Reviewers' concerns.
For convenience, we have uploaded two new Latex files, the file sensors.tex ( sensors.pdf ) includes all the revised changes without any highlighting. The file sensors.diff.tex ( sensors.diff.pdf ) makes uses of latexdiff to highlight changes between the original version and the revised manuscript. Note: there are changes in the abstract, but latexdiff cannot highlight them.Replies to Reviewers’ comments
Reviewer 1
First of all the main contribution of this paper should be outlined. It is not clear what is the main goal of this work. It seems that the paper discusses only the application of IFPS to Vehicle Networks but it does not present any new ideas or solutions.
Following the Reviewer’ concern, the manuscript has undergone a thoughtful review. All the sections have been improved, with the objective of clarifying and outlining the main goal of this work. Please find here a summary of the changes:
● The Abstract now highlights that “IPFS is a promising technology that offers a great balance between functionality, performance, and security”
● Several parts of the Introduction have been rewritten for better understanding the used technologies and remark their potential advantages over other solutions.
● The Related Work has been extended with additional citations.
● Section 2 has been expanded with additional information and a new figure.
● The Section 3 has been thoroughly revised to be clearer about how to use IPFS in Vehicular Networks. We would like to remark the importance of this section, it analyzes a technology, IPFS, that has not been designed for vehicles and RSUs.
● Sections 4 and 5 have been reviewed for English usage.
● The Results sections has been changed to emphasize the limitations and reasons because IPFS does not get excellent results but it has the potential to do it and offers better usage and security features that the state-of-the-art simulations.
● The Conclusions sections has been rewritten with the goal of clarifying and outlining the results of the investigation.
We hope that the changes done to the paper, and the arguments given in this response have addressed the Reviewer’s concern.

Reviewer 2 Report
- Introduction - "IoT devices are said to be one of the main beneficiaries of the application of semantics" - here or in Related Work you can cite the work "Using Semantic Web for Internet of Things Interoperability: A Systematic Review" - https://www.igi-global.com/article/using-semantic-web-for-internet-of-things-interoperability/210657
- Related work section is too short, I would recommend to elaborate more on listed related works or add more relevant citations
- Figure 2. shows FOAF ontology, it would be better to show excerpt from one of the mentioned vehicular network ontologies
Author Response
Dear Assistant Editor and Reviewers,
We would like to thank the Assistant Editor and the Reviewers once again for helping us to see our work from the reader's perspective and to improve its exposition accordingly. We have modified the manuscript based on the provided comments, and we hope that the modifications that we have made and the responses that we have provided herein will address the Reviewers' concerns.
For convenience, we have uploaded two new Latex files, the file sensors.tex ( sensors.pdf ) includes all the revised changes without any highlighting. The file sensors.diff.tex ( sensors.diff.pdf ) makes uses of latexdiff to highlight changes between the original version and the revised manuscript. Note: there are changes in the abstract, but latexdiff cannothighlight them.Replies to Reviewers’ comments
Reviewer 2
1) Introduction - "IoT devices are said to be one of the main beneficiaries of the application of semantics" - here or in Related Work you can cite the work "Using Semantic Web for Internet of Things Interoperability: A Systematic Review"-
https://www.igi-global.com/article/using-semantic-web-for-internet-of-things-interoperability/210657
Regarding this concern, we have revised this work and added it to the Related Work section.
2) Related work section is too short, I would recommend to elaborate more on listed related works or add more relevant citations.
Following the second concern of the Reviewer 2, the Related Work section has been extended and additional citations have been added.
3) Figure 2. shows FOAF ontology, it would be better to show excerpt from one of the mentioned vehicular network ontologies.
We completely agree with this suggestion, the relevant section has been extended and a new Figure 3 have been added. The figure shows an example using the CAOVA ontology (ontology suggested for the description for vehicle accidents).
We hope that the changes done to the paper, and the arguments given in this response have addressed the Reviewer’s concern.

Reviewer 3 Report
This paper introduces a methodology for integrating a priori knowledge with sensor data bound to a formal ontology in vehicular networks.
The paper is well written and very interesting. A good application supported with an adequate set of experimental data, and correctly related to the current literature both in sensor-based networks and in ontology. I only have a couple of concerns.
1) I do think that references to some studies referring spatial reasoning and description logic would be relevant from a related work viewpoint. I recommend
@CONFERENCE{Cristani20095,
author={Cristani, M. and Gabrielli, N.},
title={Practical issues of description logics for spatial reasoning},
year={2009},
volume={SS-09-02},
pages={5-10},
}
That looks at intelligent applications as a reference viewpoint.
2) I think that, in the introduction, authors should delimit a bit more clearly where the research bounds similar ones. In particular with respect to vehicular networks in other situations, such as cooperating robots.
For the rest: nice work.
Author Response
Dear Assistant Editor and Reviewers,
We would like to thank the Assistant Editor and the Reviewers once again for helping us to see our work from the reader's perspective and to improve its exposition accordingly. We have modified the manuscript based on the provided comments, and we hope that the modifications that we have made and the responses that we have provided herein will address the Reviewers' concerns.
For convenience, we have uploaded two new Latex files, the file sensors.tex ( sensors.pdf ) includes all the revised changes without any highlighting. The file sensors.diff.tex ( sensors.diff.pdf ) makes uses of latexdiff to highlight changes between the original version and the revised manuscript. Note: there are changes in the abstract, but latexdiff cannot highlight them.Replies to Reviewers’ comments
Reviewer 3
1) This paper introduces a methodology for integrating a priori knowledge with sensor data bound to a formal ontology in vehicular networks.
The paper is well written and very interesting. A good application supported with an adequate set of experimental data, and correctly related to the current literature both in sensor-based networks and in ontology. I only have a couple of concerns.
I do think that references to some studies referring spatial reasoning and description logic would be relevant from a related work viewpoint. I recommend “Cristani, M.; Gabrielli, N. Practical Issues of Description Logics for Spatial Reasoning. Benchmarking of Qualitative Spatial and Temporal Reasoning Systems”. That looks at intelligent applications as a reference viewpoint.
First of all, thanks for the kind words. We have tried to address the Reviewer’s concern expanding and adding new references to the Related Work section, including the suggested citation.
2) I think that, in the introduction, authors should delimit a bit more clearly where the research bounds similar ones. In particular with respect to vehicular networks in other situations, such as cooperating robots.
Regarding the Reviewer’s concern, the Introduction have been rewritten for better understanding the used technologies and remark their potential advantages over other solutions. We apologize for not comparing it appropriately with spatial reasoning and cooperating robots technologies. Our research focus is in P2P and Information-Centric Networking and we have been unable to provide relevant details about other technologies to the reader. We will keep researching those areas to include them in future works.
We hope that the changes done to the paper, and the arguments given in this response have addressed the Reviewer’s concern.

Reviewer 4 Report
The paper focuses on presenting semantic distributed data for vehicular networks using the inter-planetary file system. In this paper, the Authors proposed the creation of a semantic distributed network using content-addressed networking and peer-to-peer (P2P) connections. To meet the Authors’ aim Road-Side Units (RSUs) and vehicles applied ontologies to semantically represent information and facilitated the development of intelligent autonomous agents capable of navigate and process the shared data. The Authors used the Inter-Planetary File System (IPFS) in order to create the P2P network. According to the Authors, the conducted experiments of a real implementation of semantic sensor data using IPFS confirmed that IPFS was a straightforward way to create a reliable and secure network. The Authors tried to compare the obtained results with the other works. The topic is interesting and the paper is well corresponding to the journal aim and scope. Results are interesting and promising as the research area of semantic data representation is rapidly emerging.
While manuscript seems to be interesting in practical and experimental terms, it is difficult to point clear, methodical contribution of the paper – please focus on methodical soundness during preparing revised version of manuscript.
Please define research gap in the light of relevant and up to date studies.
The paper is well structured.
The English language needs to be improved.
Minor typos:
- Lines 13-16: “Results show that IPFS is an straightforward way …...” – please check the grammar.
- Line 138: “The Figure 2 show an example..” – please check the grammar
Author Response
Dear Assistant Editor and Reviewers,
We would like to thank the Assistant Editor and the Reviewers once again for helping us to see our work from the reader's perspective and to improve its exposition accordingly. We have modified the manuscript based on the provided comments, and we hope that the modifications that we have made and the responses that we have provided herein will address the Reviewers' concerns.
For convenience, we have uploaded two new Latex files, the file sensors.tex ( sensors.pdf )
includes all the revised changes without any highlighting. The file sensors.diff.tex
( sensors.diff.pdf ) makes uses of latexdiff to highlight changes between the original version and the revised manuscript. Note: there are changes in the abstract, but latexdiff cannot highlight them.Replies to Reviewers’ comments
Reviewer 4
1) The paper focuses on presenting semantic distributed data for vehicular networks using the inter-planetary file system. In this paper, the Authors proposed the creation of a semantic distributed network using content-addressed networking and peer-to-peer (P2P) connections. To meet the Authors’ aim Road-Side Units (RSUs) and vehicles applied ontologies to semantically represent information and facilitated the development of intelligent
autonomous agents capable of navigate and process the shared data. The Authors used the Inter-Planetary File System (IPFS) in order to create the P2P network. According to the Authors, the conducted experiments of a real implementation of semantic sensor data using IPFS confirmed that IPFS was a straightforward way to create a reliable and secure network.
The Authors tried to compare the obtained results with the other works. The topic is interesting and the paper is well corresponding to the journal aim and scope. Results are interesting and promising as the research area of semantic data representation is rapidly emerging.
While manuscript seems to be interesting in practical and experimental terms, it is difficult to point clear, methodical contribution of the paper – please focus on methodical soundness during preparing revised version of manuscript.
Please define research gap in the light of relevant and up to date studies.
Following the Reviewer’ concern, the manuscript has undergone a thoughtful review. All the sections have been improved, with the objective of clarifying and outlining the main goal of this work. The Introduction and Related Work section has been reviewed and expanded to include additional details, and we would like to remark the changes in the Results and Conclusions sections, where we have tried to add more focus adding a better explanation to the controversial results and the reasons because we consider IPFS a different, somewhat imperfect but very well-suited solution for sharing data in vehicular network.
We hope that the changes done to the paper are addressing the Reviewer’s concerns.
